# A simple and robust method for automating analysis of naïve and regenerating peripheral nerves

Alison L. Wong [1]*, Nicholas Hricz [1], Harsha Malapati[1], Nicholas von Guionneau [1], Michael Wong [2], Thomas Harris[1], Mathieu Boudreau[3], Julien Cohen-Adad [3], Sami Tuffaha[1]*

**1** Department of Plastic and Reconstructive Surgery, Johns Hopkins University, Baltimore, MD, United States of America, **2** Department of Anesthesia, Dalhousie University Faculty of Medicine, Pain Management & Perioperative Medicine, Halifax, NS, Canada, **3** NeuroPoly Lab, Institute of Biomedical Engineering, Polytechnique Montreal, Montreal, QC, Canada

* alison.wong@dal.ca (ALW); stuffah1@jhmi.edu (ST)

## Abstract

### Background

Manual axon histomorphometry (AH) is time- and resource-intensive, which has inspired many attempts at automation. However, there has been little investigation on implementation of automated programs for widespread use. Ideally such a program should be able to perform AH across imaging modalities and nerve states. AxonDeepSeg (ADS) is an open source deep learning program that has previously been validated in electron microscopy. We evaluated the robustness of ADS for peripheral nerve axonal histomorphometry in light micrographs prepared using two different methods.

### Methods

Axon histomorphometry using ADS and manual analysis (gold-standard) was performed on light micrographs of naïve or regenerating rat median nerve cross-sections prepared with either toluidine-resin or osmium-paraffin embedding protocols. The parameters of interest included axon count, axon diameter, myelin thickness, and g-ratio.

### Results

Manual and automatic ADS axon counts demonstrated good agreement in naïve nerves and moderate agreement on regenerating nerves. There were small but consistent differences in measured axon diameter, myelin thickness and g-ratio; however, absolute differences were small. Both methods appropriately identified differences between naïve and regenerating nerves. ADS was faster than manual axon analysis.

### Conclusions

Without any algorithm retraining, ADS was able to appropriately identify critical differences between naïve and regenerating nerves and work with different sample preparation

**Data Availability Statement:** All files are available from the Harvard Dataverse database (https://dataverse.harvard.edu/dataset.xhtml?persistentId=doi:10.7910/DVN/H9N9ZU).

**Funding:** The author(s) received no specific funding for this work.

**Competing interests:** We have read the journal's policy and the authors of this manuscript have the following competing interests: Mathieu Boudreau and Julien Cohen-Adad worked on the original development of AxonDeepSeg, which is an open source program. They were involved in technical support and writing but were not involved in data acquisition or analysis. Neither received or stand to receive financial compensation for this work. This does not alter our adherence to all PLOS ONE policies on sharing data and materials.

methods of peripheral nerve light micrographs. While there were differences between absolute values between manual and ADS, ADS performed consistently and required much less time. ADS is an accessible and robust tool for AH that can provide consistent analysis across protocols and nerve states.

## Introduction

Axon histomorphometry (AH) is the most commonly used outcome measure in nerve research. It involves axon quantification and measurement of axonal micro-structure parameters on nerve cross-sections [1,2]. Manual measurements have traditionally been the gold-standard for AH, but this approach is both time- and resource-intensive and has known limitations related to differing sample preparation protocols and inter-rater reliability [1,2].

Axon histomorphometry is used in studying both the central and peripheral nervous systems, but recent advances in peripheral nerve regeneration have increased its use as an outcome measure. There are now countless techniques and therapeutics being tested to improve peripheral nerve regeneration, including three-dimensional printing of nerve guides, nerve wraps, electrical stimulation, and growth hormone [3–6]. This increased interest on nerve regeneration has highlighted the need for accessible, streamlined, and robust methods of performing AH.

The recognized limitations of manual AH have inspired many tools to achieve some degree of automation [7–9]. A major obstacle to full automation using traditional programming methods is the difficulty of achieving axonal segmentation; that is, the process by which the boundaries of the myelin sheath surrounding individual axons are defined so that each axon can be measured separately [7,10,11]. To illustrate the challenge of axonal segmentation, axons in undamaged nerves reside in close proximity with myelin sheaths tending to abut each other; in comparison, debris is frequently observed in regenerating nerves that can be mislabeled as an axon. These and other nuances need to be carefully defined to improve axonal segmentation, but manual correction is still required due to frequent errors that can affect downstream calculations and substantially reduce measurement accuracy.

Machine learning has the potential to address the barriers to automating AH. Unlike the rigid design and inflexibility of traditional programs, machine learning algorithms iteratively create a set of rules by training on a large dataset [11,12]. Deep learning, also known as computer vision, is a subset of machine learning that can use convolutional neural networks, where each network hierarchically defines specific features of images and does not require structured numerical input data [13,14]. Various programs have leveraged deep learning for nerve analysis, but their generalizability has not been investigated for implementation outside of the research groups in which they were developed [8–11,14–16] and this has limited their widespread use.

AxonDeepSeg (ADS) is a novel deep learning program for fully-automated AH originally trained and validated using both scanning- and transmission-electron micrographs of the brain and spinal cord in mice, macaques, and humans [11]. Given the increased accessibility of light microscopy, and its prevalent use in the field of peripheral nerve research, the purpose of the present study was to validate the performance of ADS "out-of-the-box" in peripheral nerve light micrographs to determine if it can be implemented reliably without retraining. Given the robustness of its deep learning algorithm, we hypothesized that ADS could perform AH on light micrographs of peripheral nerves across different tissue preparation protocols and

nerve states. To test this hypothesis, we performed a validation study using adult rat median nerve preparations, comparing ADS to the gold-standard of manual axon quantification and critical parameter measurements.

## Materials and methods

### Animals

This study was carried out in accordance with National Institutes of Health recommendations in the Guide for the Care and Use of Laboratory Animals. The protocol was approved by the Johns Hopkins University Animal Care and Use Committee (Protocol number: RA18M74). All surgery was performed under isoflurane anesthesia with pre- and post-procedure subcutaneous buprenorphine analgesia. We used four adult male Lewis rats (Envigo, Frederick, MD), 12 to 24 weeks old and weighing 300 to 400 g. Two animals had no intervention prior to median nerve harvest. The other two underwent cut and repair of the proximal aspect of the median nerve at the level of the pectoralis and were sacrificed 16 weeks later. There were no specific exclusion criteria and no animals were excluded from the analysis.

### Tissue harvest

The rats were anesthetized with isoflurane prior to exposure of the heart for cardiac perfusion. Each rat was perfused through the left ventricle with 200 mL of phosphate buffered saline (PBS) followed by 200 mL of 4% paraformaldehyde (Sigma Aldrich, St Louis, MO) in PBS. Unilateral median nerves were exposed through a volar approach and dissected free of surrounding tissue under a dissection microscope. To compare embedding protocols, in the two naïve rats, 10 mm samples of each nerve were collected at the mid-humerus level. These were further cut into two 5 mm samples, each of which was allocated to one of two processing protocols: osmium-paraffin or toluidine-resin. For the regenerating samples, the median nerve was harvested at the level of the mid-forearm and nerve was processed in osmium-paraffin only.

### Tissue preparation

For this study, we compared two staining and embedding protocols as our lab commonly uses both for our peripheral nerve analysis. Toluidine blue staining and resin embedding has traditionally been used for both light and electron microscopy. This process produces very high quality micrographs, but requires a specialized set up for embedding and sectioning and can be cost prohibitive [17]. Osmium staining with paraffin embedding is a cost-effective solution that has increased accessibility in regards to time and equipment while still providing suitable image quality for analysis using light microscopy [17].

### Toluidine blue with resin embedding

One 5 mm sample from each naïve nerve was fixed for 48 hours at 4° C in a solution of 2% glutaraldehyde (16216, EMS), 3% paraformalydehyde (15754-S, EMS) and 0.1 M Sorensen's phosphate buffer pH 7.2 solution. Specimens were post-fixed in 2% osmium tetroxide, dehydrated in ascending alcohol series (starting at 50%) and embedded in Araldite® 502 resin (Polyscience). Semi-thin (1 μm) tissue sections were cut on an Ultracut E microtome (Reichert Inc, Buffalo, NY) and stained with 1% Toluidine blue.

### Osmium tetroxide staining only with paraffin embedding

This protocol is based on a previously published staining and embedding procedure with all steps carried out at 4° C [17]. One 5 mm sample from each nerve was fixed in a 4%

paraformaldehyde in PBS solution for 2 hours. Samples were then placed in two serial washes of PBS solution with gentle agitation for 30 seconds each prior to being submerged for 2 hours in 2% osmium solution (RT 19172, EMS). Thereafter, the washing step was repeated, and samples were stored in 0.2% glycine (Sigma) in PBS solution until embedding. Samples were processed by graded ethanol dehydration (starting at 30%), cleared with Pro-Par (Anatech Ltd) and infiltrated with paraffin. After processing, samples were embedded in paraffin, then cut with a microtome into 7 μm thick sections.

## Imaging

Two representative slices of naïve median nerve (one from each animal) were chosen from each of the two staining/embedding protocols (toluidine-resin, osmium-paraffin), and two representative slices of regenerating nerve (one from each animal) in osmium-paraffin were chosen. Microscopy was performed using a Zeiss Axioplan 2 (Carl Zeiss Microscopy LLC, White Plains, NY) with a 100x oil lens (numerical aperture 1.30) and digital images were captured using a Jenoptik ProgRes C5 camera (Jupiter, FL) mounted to the microscope. At this magnification, a single nerve slice produced 12 to 17 micrograph segments.

## Manual analysis

For manual analysis, the 12 to 17 non-overlapping micrograph segments for each nerve slice were analyzed individually by a blinded assessor. These micrograph segments (.tiff files, 2580x1944 pixels, 15 MB) were imported into ImageJ (FIJI Package, version 2.0, NIH, Bethesda, MD) and, per stereological principles, at the center of each micrograph segment we sampled a box measuring either 25x25 μm ($625 \ \mu m^2$; naïve nerves) or 40x40 μm ($1600 \ \mu m^2$; regenerating nerves) (Fig 1) [18]. This method created non-overlapping sample areas across the entire nerve. Axons touching the top and right borders of the box were included in the measurements while those touching the bottom and left borders were not [1]. The larger sampling area was used for regenerating nerves due to their lower axon density. If the sampled area was at an edge, and axons only covered a portion of the area, a multiplication factor was applied to ensure that the counts were comparable across all samples from the same nerve. Thresholding to create a black and white image was performed manually so that small axons were captured while simultaneously trying to avoid artificially thickening myelin sheaths. For fibers where myelin touched, lines were drawn between them so that they would be individually segmented. Counting was performed manually and then axon diameter and myelin thickness were individually measured using the "minferet" command. G-ratio (axon diameter divided by fiber diameter) was calculated from these values. "Minferet" is the minimum caliper diameter of Feret's diameter, which is the longest distance between any two points along the selection boundary, it is analogous to the minor axis of an ellipse.

## Automated analysis

AxonDeepSeg (ADS) is an open source program created using the Python coding language, and its development has been previously described [11]. In brief, deep learning models were trained from a dataset containing transmission electron micrographs with different acquisition resolutions in order to increase variability and improve generalization. All samples were from the central nervous system (brain and spinal cord) of mice, rats, and humans. The pipeline of ADS development consisted of four steps: data preparation, learning, evaluation, and prediction. Ground truth labeling for segmentation was created using the image processing software GIMP (https://www.gimp.org/) and was cross-checked by at least two researchers. The deep learning architecture is based on the U-net, which combined a contracting path with

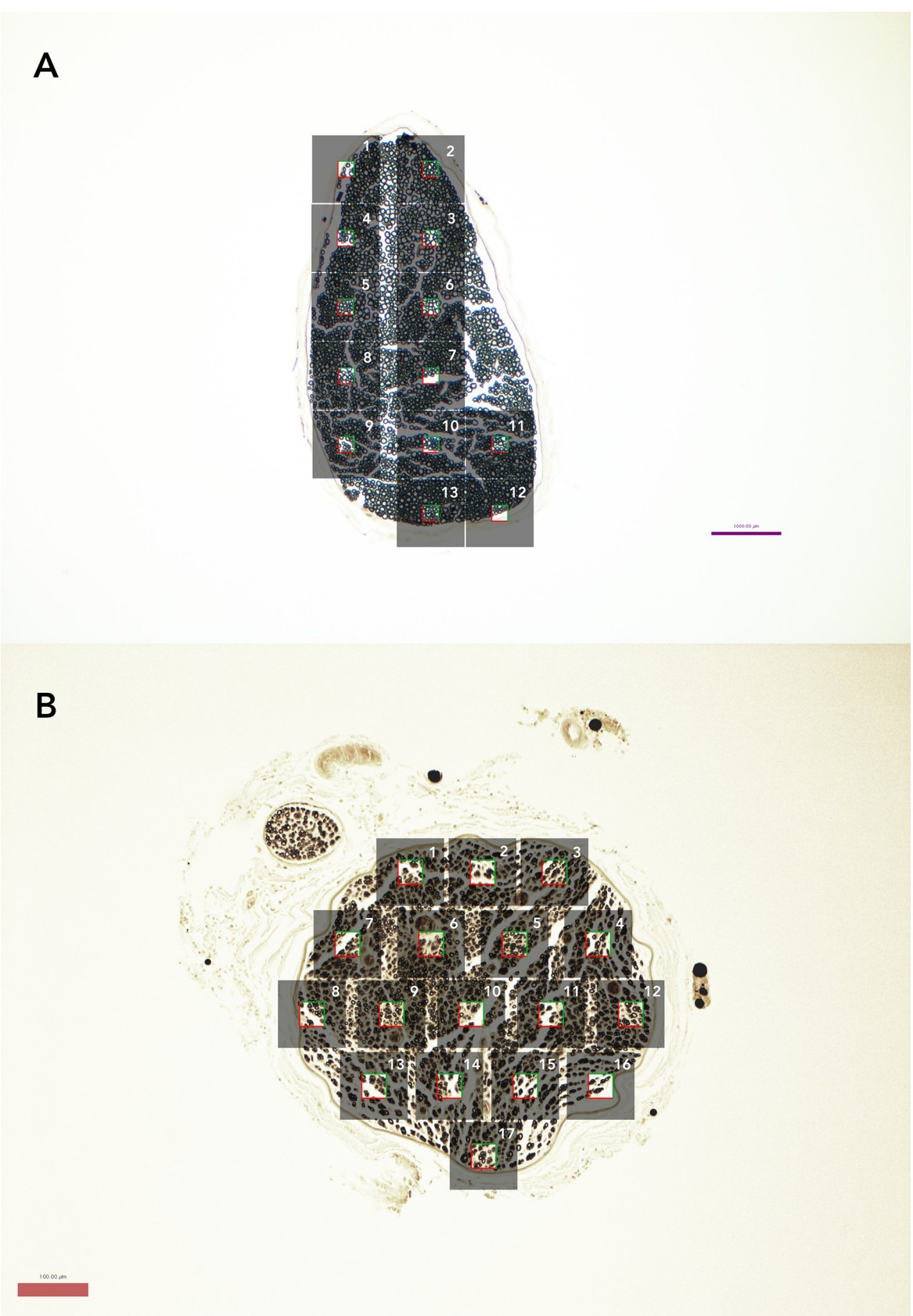

**Fig 1. Nerve slices at 10x magnification demonstrating number of micrographs and stereology sample.** (A) Representative naïve nerve (osmium-paraffin protocol). Grey boxes represent individual micrographs, while the clear square in the middle represents the stereologic sampled area of 25x25 μm box (625 μm$^2$) used for naïve samples. Axons touching the top and right borders of the box were included in the measurements while those touching the bottom and left borders were not. (B) Representative regenerating nerve (osmium-paraffin protocol), demonstrating stereologic sampling with larger 40x40 μm box (1600 μm$^2$), given decreased axon density.

traditional convolutions and then an expanding path with up-convolutions. This allowed for prediction of 3 classes: axon, myelin, and background. To avoid border effects during prediction, inference is run on a larger sample square but only calculates within a smaller square, then iterates over the entire image by shifting this sample square.

For our study, the same .tiff images used in the manual analysis were imported into ADS (version 2.1, with FSLeyes extension) and the resolution was entered (0.05 micrometer per pixel). The transmission electron microscopy pre-set was chosen because it has dark myelin and lighter axons, similar to light microscopy. Segmentation was performed automatically. Though manual correction is easily performed in the graphic user interface of ADS, to assess the accuracy of the automatic segmentation we refrained from correction beyond removal of segmented connective tissue (Fig 2). ADS then computed axon measurements for the entire micrograph and created a .csv file of each axon, its coordinates, and its respective measurements. To match the stereology sampling used in manual analysis, only axons within the same coordinates as in the manual analysis were included in our results.

## Statistical analysis

Data analyses were conducted using R (R Core Team, 3.6.1). Descriptive statistics were calculated for axon count, myelin thickness, and g-ratio. The primary outcome of comparability of axon counts was evaluated using Bland-Altman plots and intraclass correlation (ICC). The Bland-Altman plots show the difference between each value obtained from ADS and manual measurement, plotted against the mean of each measurement pair [19]; 95% of differences are expected to be contained within 1.96 standard deviations of the mean difference (i.e., the "limits of agreement"), otherwise fixed measurement bias must be ruled out. Secondary outcomes of myelin thickness, axon diameter, and g-ratio were compared using two-sample t-tests and Cohen's d for effect size. A $P$-value of $< 0.05$ was considered statistically significant. Cohen's d effect size magnitudes were defined as: negligible ($< 0.2$), small ($< 0.5$), medium ($< 0.8$), and large ($> 0.8$).

## Results

The average time for ADS to segment a 15 MB tiff file measuring 2580x1944 pixels was 20 seconds (2018 MacBook Pro, 3.1 GHz Intel Core i5, 8 GB 2133 MHz LPDDR3 RAM), with an additional minute for minimal manual correction and 1 second for measurement output. Average time to import the same image into ImageJ, perform manual thresholding, counting and measurement for a single micrograph was 5 minutes. In total there were 27 micrographs of the two toluidine-resin naïve nerves (12 and 15), 27 micrographs of the two osmium-paraffin naïve nerves (13 and 14), and 33 micrographs of the two osmium-paraffin regenerating nerves (16 and 17).

## Axon count

Average axon count in the 25x25 μm sample areas of the 27 toluidine-resin micrographs for the two naïve nerves was 12.31 ± 4.12 when counted manually and 11.79 ± 3.8 (mean ± standard deviation) in the ADS output (Fig 3A). Bland-Altman showed limits of agreement ranging from -5 to 5 and the ICC was 0.76 (P < 0.001; Fig 4A).

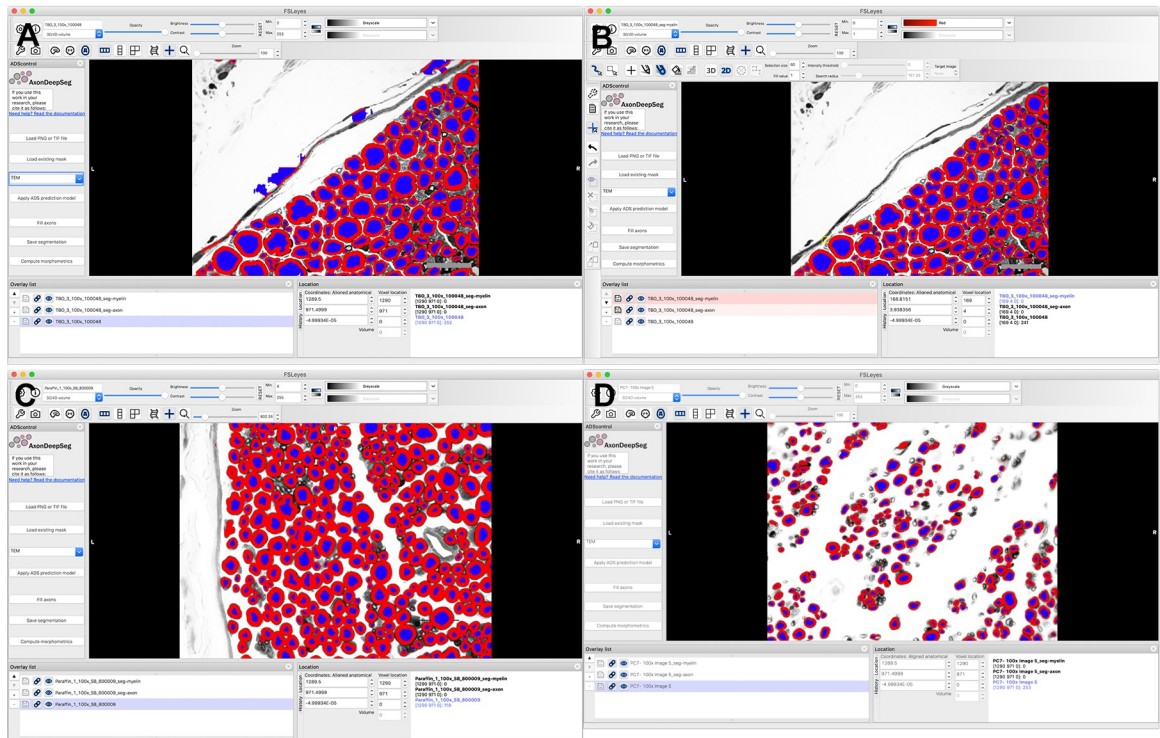

**Fig 2. Representative screen shots showing user interface of AxonDeepSeg (ADS).** (A) .tiff images were imported individually into the program, resolution was defined, and transmission electron microscopy was chosen as it produces dark myelin and lighter axons, as in light microscopy. Segmentation was then performed automatically, creating different layers for the axons and myelin. (B) Minimal manual correction was performed to remove objects mislabeled as axons. Histomorphometric measurements and axon count were then calculated by pressing the "Compute morphometrics" button, and a .csv file with these values was generated and segmented images were saved. (A) and (B) are micrographs of naïve nerves prepared in toluidine-resin protocol. (C) Micrograph of naïve nerve, osmium-paraffin protocol. (D) Micrograph of regenerating nerve, osmium-paraffin protocol.

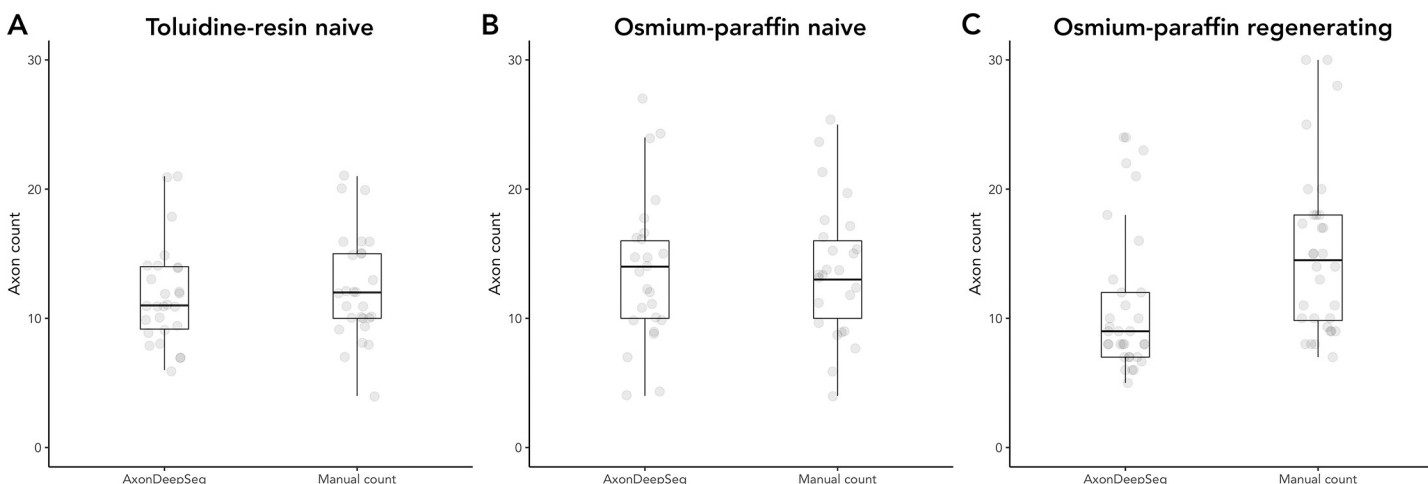

**Fig 3. Number of axons in the sampled area.** (A) Naïve nerves prepared with the toluidine-resin protocol, the mean number of axons in the 625 μm$^2$ sample area for the 27 micrographs. (B) Naïve nerves prepared with the osmium-paraffin protocol, the mean number of axons in the 625 μm$^2$ sample area for the 27 micrographs. (C) Regenerating nerves prepared with the osmium-paraffin protocol, the mean number of axons in the 1600 μm$^2$ sample area for the 33 micrographs. Boxes are 1 interquartile range (IQR), whiskers 1.5*IQR.

Axon count in the same sized sample area for the 27 micrographs osmium-paraffin protocol naïve nerves was 13.72 ± 5.19 when counted manually, and was 13.72 ± 5.78 in the ADS output (Fig 3B). Bland-Altman comparison showed good agreement between the counts, with limits of agreement ranging from -6 to 6. ICC was 0.84 (P < 0.001), indicating good to excellent agreement (Fig 4B). In the regenerating nerves, the average axon count per 40x40 μm sample area was 15.96 ± 7.48 when counted manually and 11.09 ± 5.74 in the ADS output (Fig 3C). Bland-Altman limits of agreement were -2 to 12 and the ICC was 0.56 (P <0.001; Fig 4C).

## Measurements

In the naïve nerve toluidine-resin protocol samples, there was a small difference in measured axon diameters, with larger measurements in ADS compared to manual (3.67 ± 1.92 vs. 3.06 ± 1.71, t = 4.19, P < 0.001; d = 0.34; Fig 5A). Myelin was thicker in the ADS compared to manual measuring (1.16 ± 0.52 vs. 1.06 ± 0.46, t = 2.63, P = 0.0089; d = 0.21; Fig 6A). Thus g-ratio (axon diameter divided by fiber diameter) was larger in ADS (0.59 ± 0.12 vs. 0.57 ± 0.11, t = -2.22, P = 0.027; d = 0.18; Fig 7A).

In naïve nerve osmium-paraffin protocol samples, there was no difference in measured axon diameter measured by ADS or manually (2.45 ± 1.16 vs. 2.47 ± 1.13, t = -0.29, P = 0.78; Fig 5B). Myelin was measured as thicker in ADS compared to manual measurements (1.54 ± 0.45 vs. 1.35 ± 0.54, t = 4.86, P < 0.001; d = 0.38; Fig 6B), and this difference was consistent across micrographs. In turn, ADS had a smaller g-ratio, defined as axon diameter divided by fiber diameter (0.425 ± 0.10 vs. 0.474 ± 0.10, t = -6.28, P < 0.001; d = 0.49; Fig 7B).

In regenerating nerves (osmium-paraffin protocol), axon diameters were notably smaller than in the naïve nerves in both ADS (regenerating vs. naïve: 1.84 ± 1.00 vs. 2.45 ± 1.16, t = 7.13, P < 0.001, d = 0.56) and manual measurements (1.27 ± 0.87 vs. 2.47 ± 1.13, t = 16.3, P <0.001, d = 1.23). ADS measured axon diameter as larger compared to manual measurements (1.84 ± 1.00 vs. 1.27 ± 0.87, t = 8.46, P < 0.001; d = 0.62; Fig 5C). Unlike in naïve samples, ADS measured myelin thickness as less than manual measuring (0.86 ± 0.30 vs. 1.06 ± 0.32, t = -8.56, P < 0.001; d = -0.60; Fig 6C). This created a significantly larger calculated g-ratio in the ADS data (0.50 ± 0.16 vs. 0.39 ± 0.14, t = 13.8, P < 0.001; d = 1.0; Fig 7C).

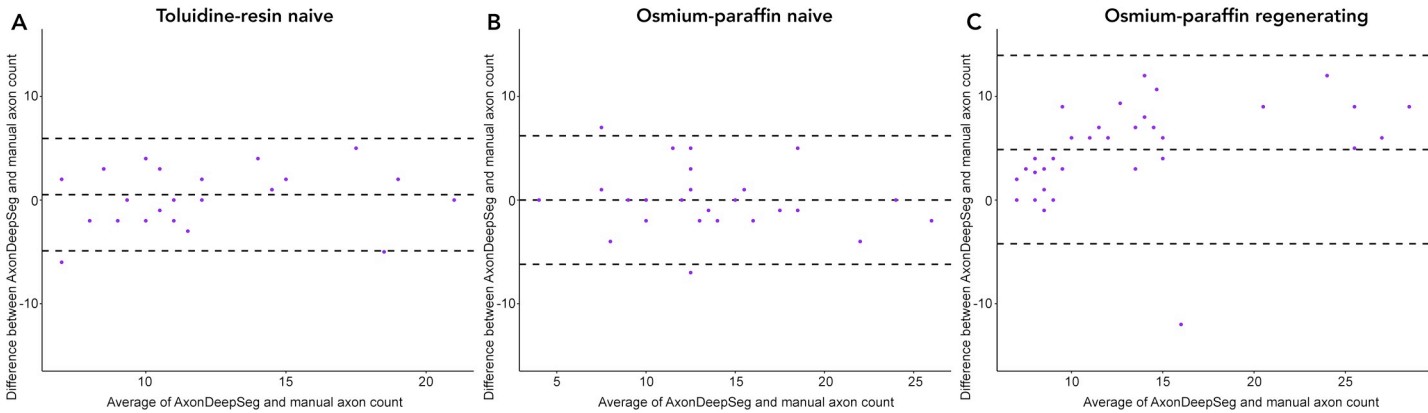

**Fig 4. Bland-Altman plots of ADS and manual axon counts.** (A) In naïve nerves prepared with the toluidine-resin protocol, the limits of agreement between ADS and manual counts in the 625 μm² sample area in each micrograph ranged from -5 to 5. (B) In naïve nerves prepared with the osmium-paraffin protocol, the limits of agreement between ADS and manual counts in the 625 μm² sample area ranged from -6 to 6. (C) In regenerating nerves prepared with the osmium-paraffin protocol, the limits of agreement between ADS and manual counts in the 1600 μm² sample area ranged from -2 to 12.

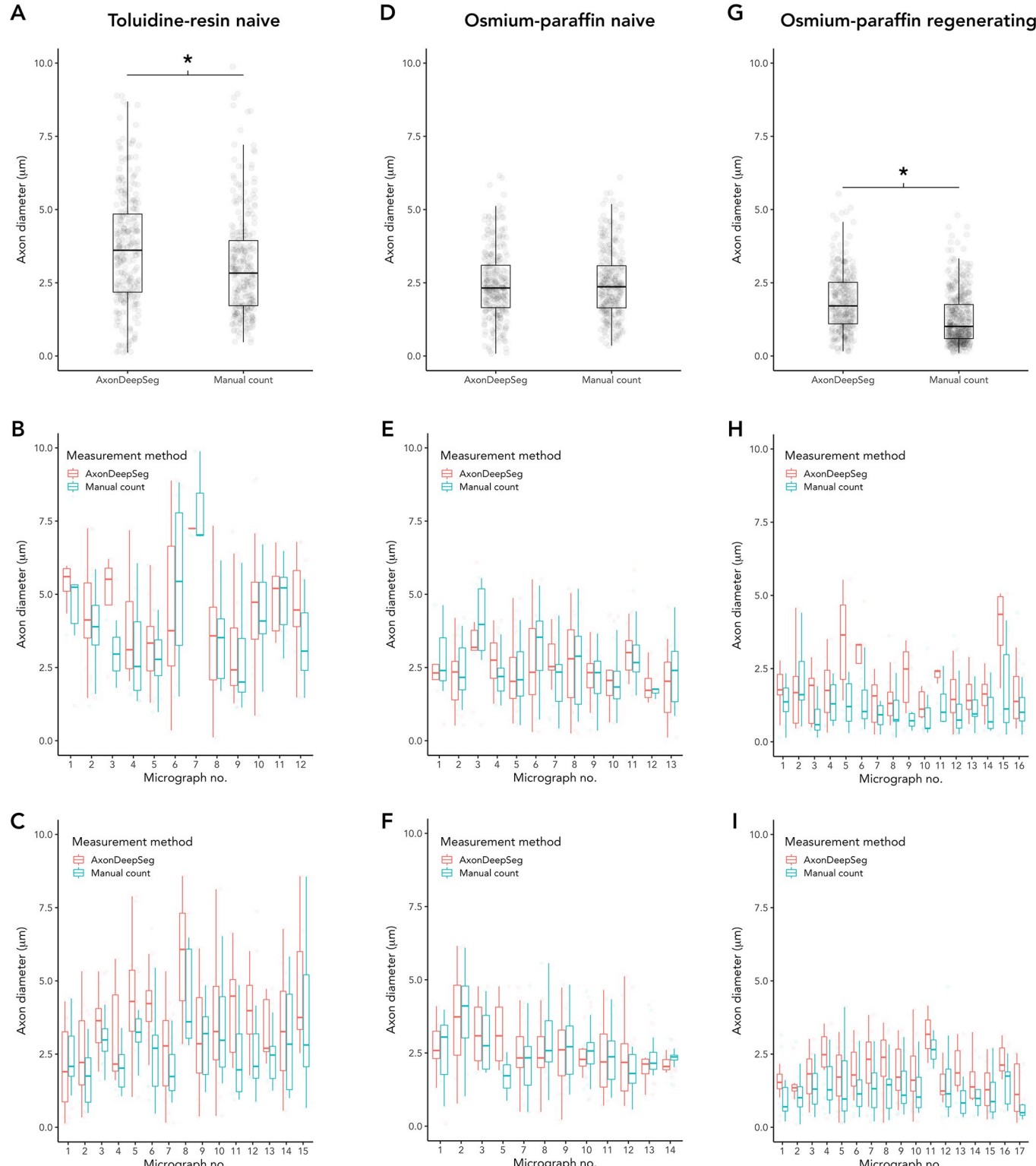

**Fig 5. Comparison of axon diameters measured by ADS or manually.** (A-C) Naïve nerves, toluidine-resin protocol; (A) Mean axon diameters for all micrographs for both nerves; (B) Mean axon diameters in each micrograph for nerve one; (C) Mean axon diameters in each micrograph for nerve two. (D-F) Naïve nerves, osmium-resin protocol; (D) Mean axon diameters for all micrographs for both nerves; (E) Mean axon diameters in each micrograph for nerve one; (F) Mean axon diameters in each micrograph for nerve two. (G-I) Regenerating nerves, osmium-paraffin protocol; (G) Mean axon diameters for all micrographs for both nerves; (H) Mean axon diameters in each micrograph for nerve one; (I) Mean axon diameters in each micrograph for nerve two. Asterisk (*) = significant (P < 0.05 by two-sample t-test). Boxes are 1 interquartile range (IQR), whiskers 1.5*IQR.

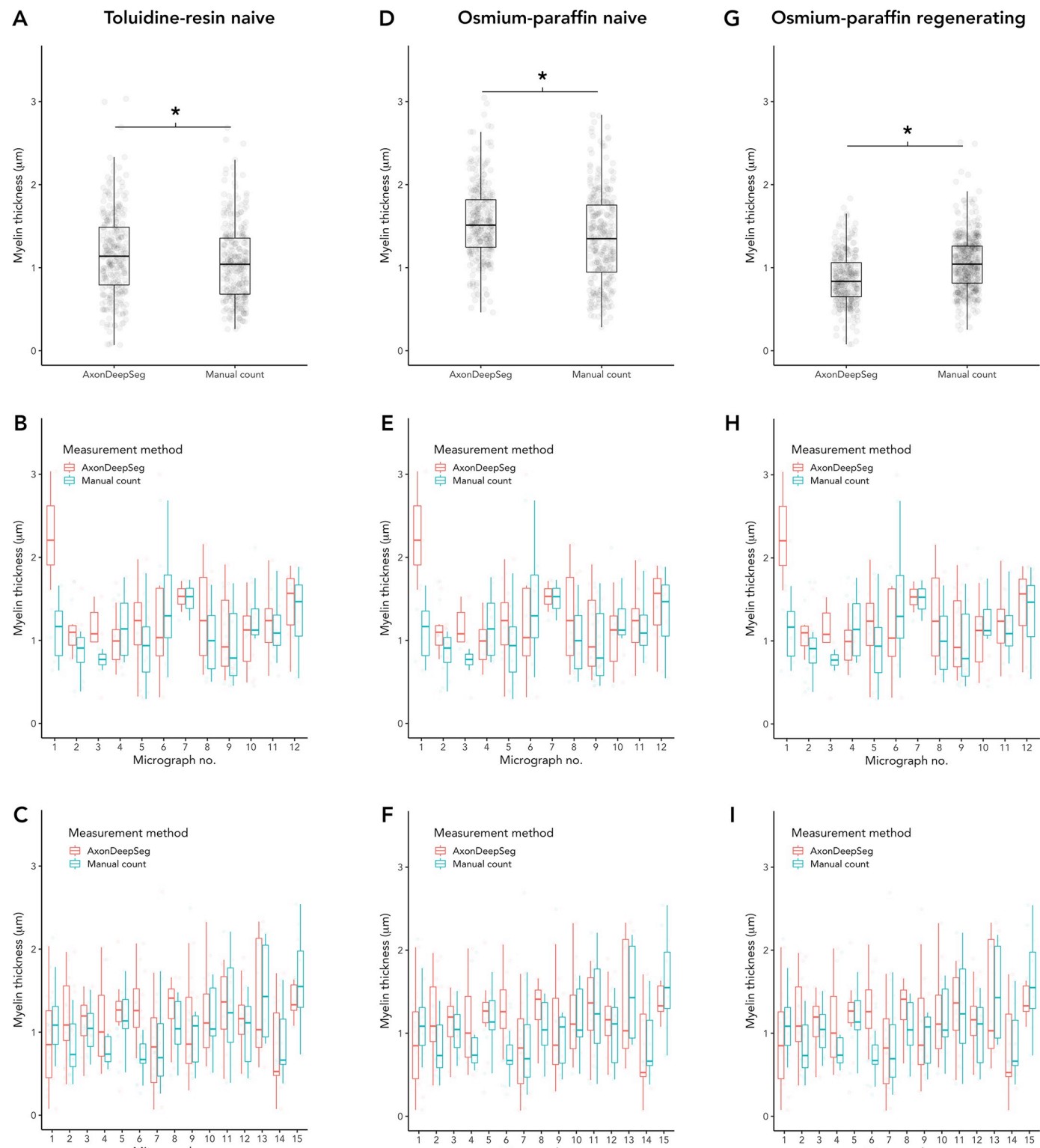

**Fig 6. Comparison of myelin thickness measured by ADS or manually.** (A-C) Naïve nerves, toluidine-resin protocol; (A) Mean myelin thickness for all micrographs for both nerves; (B) Mean myelin thickness in each micrograph for nerve one; (C) Mean myelin thickness in each micrograph for nerve two. (D-F) Naïve nerves, osmium-resin protocol; (D) Mean myelin thickness for all micrographs for both nerves; (E) Mean myelin thickness in each micrograph for nerve one; (F) Mean myelin thickness in each micrograph for nerve two. (G-I) Regenerating nerves, osmium-paraffin protocol; (G) Mean myelin thickness for all micrographs for both nerves; (H) Mean myelin thickness in each micrograph for nerve one; (I) Mean myelin thickness in each micrograph for nerve two. Asterisk (*) = significant (P < 0.05 by two-sample t-test). Boxes are 1 interquartile range (IQR), whiskers 1.5*IQR.

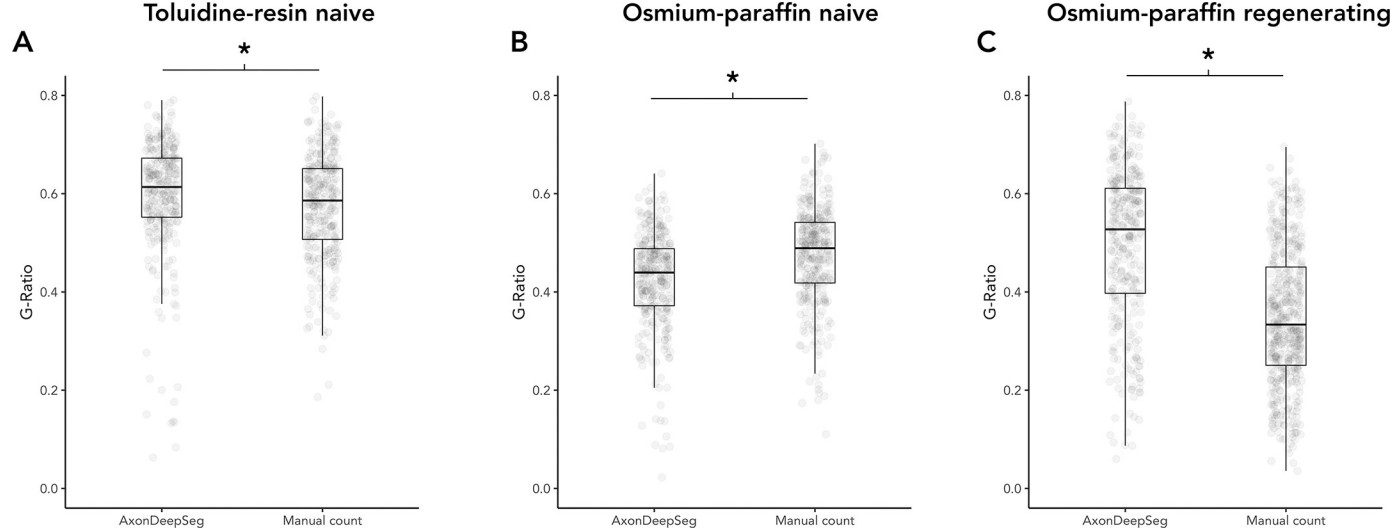

**Fig 7. Comparison of g-ratio measured by ADS or manually.** (A) Naïve nerves, toluidine-resin protocol, mean g-ratio for all micrographs for both nerves. (B) Naïve nerves, osmium-paraffin protocol, mean g-ratio for all micrographs for both nerves. (C) Regenerating nerves, osmium-paraffin protocol, mean g-ratio for all micrographs for both nerves. Asterisk (*) = significant (P < 0.05 by two-sample t-test). Boxes are 1 interquartile range (IQR), whiskers 1.5*IQR.

## Discussion

We aimed to validate the use of the newly developed deep learning program AxonDeepSeg (ADS) for automated axon histomorphometry (AH) and found that even without algorithm retraining, ADS was able to appropriately identify critical differences between naïve and regenerating nerves and work in different sample preparation methods. AH is one of the most important measures in peripheral nerve research, yet it is a time- and resource-intensive task, which limits its accessibility as an outcome measure. While there has been a boom in programs created for AH, none of these have been validated for use outside of the specific protocols for which they were developed and therefore widespread implementation has been poor.

The ADS deep learning algorithm was originally trained and validated using electron micrographs, but given the increased accessibility of light microscopy, we tested its application in light micrographs of peripheral nerves [11]. We found that ADS was able to perform near automatic AH of peripheral nerve light micrographs with results that correlated well to manual analysis but was much faster. Axon quantification in naïve nerves in both sample preparation protocols showed very good agreement with good interclass correlation between the manual and ADS counts. In regenerating nerves, the agreement was moderate, with ADS consistently counting fewer axons. Though ADS allows for manual correction of, we refrained from significant correction apart from removal of obvious connective tissue. More rigorous correction likely would have improved axon counts as it would have further minimized segmentation errors but it would be unlikely to change measurements (i.e. diameter, myelin thickness) as these were consistent within micrographs.

One of the key differences between ADS and manual axon quantification is that ADS counted and measured every axon within a micrograph. With our manual analysis we used a stereologic sampling approach where only axons within a 25x25 μm (naïve) or 40x40 μm (regenerating) box at the center of each micrograph segment was counted, resulting in 12–17 sampled areas for each nerve. This approach is widely used in AH, in order to expedite time-consuming manual measurements (REF). Using a stereologic sampling approach, however, increases the risk of sampling error and bias as a difference in that small sample area can get

compounded, leading to a significant difference in the number of total axons. For our comparisons we specifically analyzed only the ADS output for the same coordinates as in our manual analysis; however, ADS can evaluate the entirety of each micrograph rapidly which negates the need for stereologic sampling. It therefore does not need to be limited by local variation in axon density, and can avoid the error and bias propagation inherent in extrapolating from a stereologic approach.

Axon diameter, myelin thickness, and g-ratio are important AH parameters as they differentiate subpopulations of axons and can be used for evaluating changes following denervation and reinnervation [20]. We found that ADS was able to appropriately identify differences in axon diameter, myelin thickness and g-ratio between naïve and regenerating nerves. Though there were small differences between ADS and manual measurements, they were in a consistent direction. With our large dataset, this resulted in statistically significant differences even though the absolute differences were small. The absolute values of these parameters are known to differ between research labs due to differences in staining and imaging protocols [9,10], and the judgment of the individual raters doing the segmentations. Therefore, consistency within studies is most important, and ADS was able to provide reliable measurements with comparable variability as manual analysis.

Conventional machine learning has been applied to axon histomorphometry in the past in the form of random forest and bootstrapping methods, yet such techniques were still not robust enough to handle variations in these data [8,9,21]. More recently, other research groups have started applying deep learning to axon histomorphometry, these all follow the basic tenants of developing a deep learning program but differ in their training datasets and network structure. Naito et al. [10] developed a deep learning algorithm using a database of light micrographs of diseased human sural nerves. Their algorithm was trained using these light micrographs but was not tested outside of their initial database. Janjic et al. [15] developed a segmentation program using electron micrographs of human prefrontal cortex. While both of these programs showed excellent agreement when compared to manual analysis of micrographs from the same dataset, the robustness of the algorithms was not tested.

A known limitation of all machine learning programs is that accuracy and robustness are dependent on the images used to develop the model, what is known as the "ground truth". This presents a problem for AH since as previously discussed, micrographs can vary depending on nerve state and sample processing. Re-training an algorithm model can provide more accurate results, at the cost of having to re-train a model, which is time consuming, requires expertise, and still lacks generalizability [22].

Besides leveraging deep learning, ADS has a number of advantages over other programs that have been developed for axon histomorphometry [9–11,23]. ADS is open source and freely available for download from GitHub (https://github.com/neuropoly/axondeepseg). It was developed using Python, a commonly used programming language, and remains under continuous active development more than 2 years after its initial release. It has a graphical user interface and does not require advanced programming knowledge to use. Our results demonstrate that the current ADS algorithm is very robust across multiple nerve states and sample processing protocols.

Deep learning and computer vision programs are an important step forward in the analysis of outcomes measures in nerve research. The speed and reliability of these methods promise to be substantially improved over traditional manual measurement. Our work shows that in its current iteration, ADS is an accessible and robust tool for peripheral nerve AH across multiple image preparation modalities without retraining. In the future we hope to further test ADS's ability to analysis non-myelinated axons for sensory nerve recovery.

## Author Contributions

**Conceptualization:** Alison L. Wong, Nicholas von Guionneau, Thomas Harris.

**Data curation:** Alison L. Wong, Nicholas Hricz, Harsha Malapati, Nicholas von Guionneau, Thomas Harris.

**Formal analysis:** Alison L. Wong, Nicholas Hricz, Michael Wong.

**Investigation:** Nicholas Hricz, Harsha Malapati, Nicholas von Guionneau, Thomas Harris.

**Methodology:** Alison L. Wong, Harsha Malapati, Nicholas von Guionneau, Thomas Harris.

**Project administration:** Sami Tuffaha.

**Resources:** Mathieu Boudreau, Julien Cohen-Adad.

**Software:** Mathieu Boudreau, Julien Cohen-Adad.

**Supervision:** Sami Tuffaha.

**Visualization:** Alison L. Wong.

**Writing – original draft:** Alison L. Wong, Michael Wong.

**Writing – review & editing:** Alison L. Wong, Nicholas Hricz, Harsha Malapati, Nicholas von Guionneau, Michael Wong, Thomas Harris, Mathieu Boudreau, Julien Cohen-Adad, Sami Tuffaha.

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
