## [Decision Letter · Decision Letter 0]

27 Apr 2021

PONE-D-21-05849

A simple and robust method for automating analysis of naïve and regenerating peripheral nerves

PLOS ONE

Dear Dr. Wong,

Thank you for submitting your manuscript to PLOS ONE. After careful consideration, we feel that it has merit but does not fully meet PLOS ONE’s publication criteria as it currently stands. Therefore, we invite you to submit a revised version of the manuscript that addresses the points raised during the review process.

The manuscript is appreciated by all reviewers, but minor revisions are requested.

Did the authors note any differences between embedding protocols? Resin vs. paraffin. This would be interesting to know as this has implications on time commitment to sample preparation.

Have the authors attempted to manually correct the ADS data to determine if this would further improve the accuracy? As ADS much faster than manual segmentation, it may be beneficial to include this step if this is not a time consuming process.

The discussion would be improved if this methodology is compared with other published reports. This is not to say the authors should implement other methodologies but compare other authors result with their own.

Can the authors comment on the efficacy of counting non-myelinated axons. This is the bigger challenge and is important when considering sensory nerve recovery.

The quality of the figures is quite low; please improve the resolution of the figures showing the data (Figures 1/2 appear fine).

Please, consider moving your motivation for using ADS to the introduction, rather than the discussion, so the reader understands up front why you are studying this software in particular.

Please, consider introduction of the g-ratio definition early.

Line 77: Period missing.

Line 175: Can you explain the ‘minferet’ command for users not familiar with imagej?

Line 196: Does the GIMP software automatically label these images? Any reason why the software was not compared using this software?

Line 362 a citation has not been included.

Line 363: Ref not inserted

Please, provide in the INTRODUCTION examples of recent works on peripheral nerve regeneration (see, for example doi: 10.1088/1758-5090/aaaf50).

We look forward to receiving your revised manuscript.

Kind regards,

Gennady S. Cymbalyuk, Ph.D.

Academic Editor

PLOS ONE

Journal Requirements:

'I have read the journal's policy and the authors of this manuscript have the following competing interests: Mathieu Boudreau and Julien Cohen-Adad worked on the original development of AxonDeepSeg, which is an open source program. They were involved in technical support and writing but were not involved in data acquisition or analysis. Neither received or stand to receive financial compensation for this work.'

a. Please confirm that this does not alter your adherence to all PLOS ONE policies on sharing data and materials, by including the following statement: "This does not alter our adherence to  PLOS ONE policies on sharing data and materials.” (as detailed online in our guide for authors http://journals.plos.org/plosone/s/competing-interests).  If there are restrictions on sharing of data and/or materials, please state these.

Please note that we cannot proceed with consideration of your article until this information has been declared.

Reviewers' comments:

Reviewer's Responses to Questions

**Comments to the Author**

1. Is the manuscript technically sound, and do the data support the conclusions?

Reviewer #1: Yes

Reviewer #2: Yes

Reviewer #3: Yes

2. Has the statistical analysis been performed appropriately and rigorously? 

Reviewer #1: Yes

Reviewer #2: Yes

Reviewer #3: Yes

3. Have the authors made all data underlying the findings in their manuscript fully available?

Reviewer #1: Yes

Reviewer #2: No

Reviewer #3: Yes

4. Is the manuscript presented in an intelligible fashion and written in standard English?

Reviewer #1: Yes

Reviewer #2: Yes

Reviewer #3: Yes

5. Review Comments to the Author

Reviewer #1: The author's are to be congratulated on the quality of the manuscript presented and for addressing a key component that has limited peripheral nerve research. The application of ADS is novel as it is free and provides fast results and is of great interest to the research community. However, there are a few issues that need to be addressed prior to acceptance.

1) Did the authors note any differences between embedding protocols? Resin vs. paraffin. This would be interesting to know as this has implications on time commitment to sample preparation.

2) Have the authors attempted to manually correct the ADS data to determine if this would further improve the accuracy? As ADS much faster than manual segmentation, it may be beneficial to include this step if this is not a time consuming process

3) The discussion would be improved if this methodology is compared with other published reports. This is not to say the authors should implement other methodologies but compare other authors result with their own.

4) Can the authors comment on the efficacy of counting non-myelinated axons. This is the bigger challenge and is important when considering sensory nerve recovery.

5) In line 362 a citation has not been included.

Reviewer #2: Manual axon histomorphometry (AH) is time- and resource-intensive, which has inspired

many attempts at automation. Axon histomorphometry using ADS and manual analysis (gold-standard) was performed on light micrographs of naïve or regenerating rat median nerve cross-sections prepared with

either toluidine-resin or osmium-paraffin embedding protocols. The parameters of interest included axon count, axon diameter, myelin thickness, and g-ratio. In the current study, the authors aimed to validate the use of the newly developed deep learning program AxonDeepSeg (ADS) for automated axon histomorphometry (AH) and found that even without algorithm retraining, ADS was able to appropriately identify critical differences between naïve and regenerating nerves and work in different sample preparation methods. This was a carefully done counting procedure. However, in the INTRODUCTION the authors shall give several examples of recent works on peripheral nerve regeneration (see, for example doi: 10.1088/1758-5090/aaaf50)

Reviewer #3: Reviewer Comments

This manuscript presents results using an automated deep learning software, AxonDeepSeg, to automatically perform axon histomorphometry. The automated results are compared to the gold standard hand count method, and the results between the two are compared. The main outcomes are measuring axon size/diameter, myelin thickness, and the g-ratio. The results demonstrate that the automated method can detect differences between naïve and regenerating axons, in terms of axon density and g-ratio, although some differences between this method and the hand counts.

This article is straightforward and the results are well documented. Automated methods for axon histomorphometry are an important tool that need to be widely developed/deployed, and this article helps demonstrate the utility of one of the proposed solutions that is already available. The article could be made more impactful if several different automated methods were compared head-to-head in order to benchmark the currently available tools, but this manuscript still has validity by presenting a single software against manual counts. I have a few comments that should be addressed before publication, however, which are mostly minor.

Minor Comments

General: The quality of the figures is quite low; please improve the resolution of the figures showing the data (Figures 1/2 appear fine).

General: I would recommend moving your motivation for using ADS to the introduction, rather than the discussion, so the reader understands up front why you are studying this software in particular.

General: You may want to introduce the g-ratio definition early, I do not think it is explained before discussing in the results.

Line 77: Period missing.

Line 175: Can you explain the ‘minferet’ command for users not familiar with imagej?

Line 196: Does the GIMP software automatically label these images? Any reason why the software was not compared using this software?

Line 363: Ref not inserted

6. PLOS authors have the option to publish the peer review history of their article (what does this mean?). If published, this will include your full peer review and any attached files.

Reviewer #1: **Yes: **Jonatha I. Leckenby

Reviewer #2: No

Reviewer #3: No

---

## [Author Response · Author response to Decision Letter 0]

28 May 2021

Thank you for the opportunity to submit our revised manuscript, “A simple and robust method for automating analysis of naïve and regenerating peripheral nerves.” We believe that the comments and suggestions have significantly improved the manuscript and hope that you find the changes satisfactory. Below are the itemized responses to the reviewers along with the corresponding line numbers where applicable.

Did the authors note any differences between embedding protocols? Resin vs. paraffin. This would be interesting to know as this has implications on time commitment to sample preparation.

Our lab has been using both embedding protocols due to the increased accessibility and decreased costs of paraffin embedding, we have also found the paraffin protocol to be somewhat faster. A statement to this effect has been added to the methods section (Line 152).

Have the authors attempted to manually correct the ADS data to determine if this would further improve the accuracy? As ADS much faster than manual segmentation, it may be beneficial to include this step if this is not a time consuming process.

We tried to only do minimal corrections. More in-depth corrections likely would have improved the accuracy of axon counts, but we do not think that it would have changed the differences seen in the measurements of axon diameter or myelin thickness. This was mentioned in our methods but has also been added to our discussion (Line 329).

The discussion would be improved if this methodology is compared with other published reports. This is not to say the authors should implement other methodologies but compare other authors result with their own.

More details around the methodology used in other reports has been added. Briefly it was combined with the rationale behind using ADS in the introduction (Line 103), and more thoroughly methodology is now compared in the discussion (Line 435).

Can the authors comment on the efficacy of counting non-myelinated axons. This is the bigger challenge and is important when considering sensory nerve recovery.

We completely agree that there is an important role of the analysis of non-myelinated axons for sensory nerves. For this study, we focused on light microscopy, where the common methods of tissue preparation and staining focus on staining the myelin itself. Sensory nerves typically are analyzed using electron microscopy. ADS was developed in electron micrographs and likely could be used for the analysis for unmyelinated axons, but this has not yet been tested. We have added this to the conclusion as a future area of interest (Line 468).

The quality of the figures is quite low; please improve the resolution of the figures showing the data (Figures 1/2 appear fine).

We are not sure why this is the case and have now verified them using the PACE figure application so the resolution should not be a problem.

Please, consider moving your motivation for using ADS to the introduction, rather than the discussion, so the reader understands up front why you are studying this software in particular.

Thank you for the suggestion, this has been done (Line 103 and 110).

Please, consider introduction of the g-ratio definition early.

The definition of g-ratio is present in the description of the methodology used in the manual analysis (Line 214), the definition has now also been added to the “Measurements” section of the results for increased clarity (Line 325).

Line 77: Period missing.

This has been added.

Line 175: Can you explain the ‘minferet’ command for users not familiar with imagej?

This has been added (Line 215).

Line 196: Does the GIMP software automatically label these images? Any reason why the software was not compared using this software?

I am not familiar with using GIMP for labeling images. I know that in GIMP or Photoshop images can be overlaid to visually assess differences in the images. The larger issue is that with Fiji/ImageJ, the thresholded images are not automatically saved, so we could not compare the graphic outputs of each software program (ImageJ vs. ADS).

Line 362 a citation has not been included.

This sentence was referring back to our results, and therefore a citation was not included. The sentence itself has been changed to make this more clear. “We found that ADS was able to appropriately identify differences in axon diameter, myelin thickness and g-ratio between naïve and regenerating nerves.” (now found on line 414)

Line 363: Ref not inserted

Line 363 is referring to findings, perhaps the reviewer meant line 373? An appropriate citation has been added to that statement to provide support (Line 453).

Please, provide in the INTRODUCTION examples of recent works on peripheral nerve regeneration (see, for example doi: 10.1088/1758-5090/aaaf50).

A brief overview of some of the advances in peripheral nerve regeneration have been added to the introduction (Line 72).

Thank you for your consideration,

Alison Wong, MD, MSE, FRCS(C)

---

## [Decision Letter · Decision Letter 1]

16 Jun 2021

A simple and robust method for automating analysis of naïve and regenerating peripheral nerves

PONE-D-21-05849R1

Dear Dr. Wong,

We’re pleased to inform you that your manuscript has been judged scientifically suitable for publication and will be formally accepted for publication once it meets all outstanding technical requirements.

Kind regards,

Gennady S. Cymbalyuk, Ph.D.

Academic Editor

PLOS ONE

Additional Editor Comments (optional):

Reviewers' comments:

Reviewer's Responses to Questions

**Comments to the Author**

1. If the authors have adequately addressed your comments raised in a previous round of review and you feel that this manuscript is now acceptable for publication, you may indicate that here to bypass the “Comments to the Author” section, enter your conflict of interest statement in the “Confidential to Editor” section, and submit your "Accept" recommendation.

Reviewer #1: All comments have been addressed

Reviewer #2: All comments have been addressed

Reviewer #3: All comments have been addressed

2. Is the manuscript technically sound, and do the data support the conclusions?

Reviewer #1: Yes

Reviewer #2: Yes

Reviewer #3: Yes

3. Has the statistical analysis been performed appropriately and rigorously? 

Reviewer #1: Yes

Reviewer #2: Yes

Reviewer #3: Yes

4. Have the authors made all data underlying the findings in their manuscript fully available?

Reviewer #1: Yes

Reviewer #2: Yes

Reviewer #3: Yes

5. Is the manuscript presented in an intelligible fashion and written in standard English?

Reviewer #1: Yes

Reviewer #2: Yes

Reviewer #3: Yes

6. Review Comments to the Author

Reviewer #1: The Authors present an improved revision of their manuscript. All my questions or comments have been addressed in a satisfactory manner and the manuscript is acceptable for publication.

Reviewer #2: The authors have satisfactorily addressed my concerns, the manuscript has been improved and could be published in its present form

Reviewer #3: The figures (3-7) still appear to be low quality. This may be an issue with the downloaded PDF, but I will let the authors and the editorial staff address this issue moving forward. All other issues have been addressed.

7. PLOS authors have the option to publish the peer review history of their article (what does this mean?). If published, this will include your full peer review and any attached files.

Reviewer #1: **Yes: **Jonathan I. Leckenby, MBBS, PhD

Reviewer #2: No

Reviewer #3: **Yes: **Thomas Eggers

---

## [Editor Report · Acceptance letter]

24 Jun 2021

PONE-D-21-05849R1 

A simple and robust method for automating analysis of naïve and regenerating peripheral nerves 

Dear Dr. Wong:

I'm pleased to inform you that your manuscript has been deemed suitable for publication in PLOS ONE. Congratulations! Your manuscript is now with our production department. 

Kind regards, 

on behalf of

Dr. Gennady S. Cymbalyuk 

Academic Editor

PLOS ONE